

# SSM-FastICANet: a hybrid state space and FastICA model for economic growth forecasting in energy-economy-environment systems

Fahman Saeed

Computer Science Department, College of Computer and Information Sciences, Imam Mohammad Ibn Saud Islamic University (IMSIU), Riyadh, Saudi Arabia

## ABSTRACT

This study examines the complex interactions between $CO_2$ emissions, economic growth, and energy consumption across various classifications of countries. In this study, we propose SSM-FastICANet, a novel hybrid model that integrates state space models with independent component analysis and a diagonal structure for efficient and accurate economic growth forecasting a predictive model that can forecast economic growth by analyzing energy consumption patterns and emission levels, while also pinpointing the distinct impacts of $CO_2$ emissions and energy usage. Employing a time-series dataset and an innovative hybrid methodology that combines state space models (SSMs) with fast independent component analysis (FastICA), the study reveals unique interaction patterns among these variables. The FastICA method aids in uncovering essential underlying patterns and reducing dimensionality, whereas the SSM architecture proficiently captures temporal dependencies and emphasizes the most pertinent input features for precise prediction and impact detection. The model utilizes entropy, kurtosis, and variance to filter independent components, guaranteeing that the chosen features are statistically significant, locally structured, and resilient to noise. The findings demonstrate that SSM-FastICANet significantly enhances feature selection, model adaptability, and interpretability, yielding reliable predictions over various time intervals. It enhances the comprehension of the energy-economy-environment relationship and offers a solid framework for policymakers to develop strategies that foster sustainable economic growth while reducing environmental impact.

# INTRODUCTION

Forecasting economic growth within the energy-economy-environment (EEE) nexus poses numerous significant issues. The interrelations between energy consumption, $CO_2$ emissions, and economic development are markedly nonlinear and fluctuate according to temporal, geographical, and policy contexts (*Han et al., 2024*). Secondly, conventional econometric models find it challenging to account for temporal dependencies and

Corresponding author
Fahman Saeed,
faesaeed@imamu.edu.sa

multi-scale interactions, frequently resulting in oversimplified results (*Verburg, Eickhout & van Meijl, 2008*). Third, annual data—particularly in developing countries—often exhibits noise and incompleteness, so confounding pattern discovery (*Mukherjee, White & Wuyts, 1998*). Ultimately, current models frequently lack the ability to dynamically adjust to varying input conditions or to include the interpretability required for practical policy suggestions. These constraints highlight the necessity for a model that is both flexible and comprehensible across diverse scales and contexts.

Artificial intelligence (AI) serves as an essential component in tackling the intricate relationship between energy, economy, and environment. This approach connects economic development with environmental sustainability through precise predictsions, practical insights, and enhanced strategies (*Ukoba et al., 2024*). In a world confronting pressing climate challenges, AI-driven strategies empower policymakers, businesses, and scholars to make informed decisions that reconcile immediate economic objectives with enduring environmental health. Utilizing AI allows us to progress toward a future in which economic growth is separated from environmental harm, promoting sustainable development for future generations.

This investigation seeks to predict upcoming economic growth by examining the impact of energy usage and carbon dioxide emissions. Employing a sophisticated dynamic state space model (SSM) (*Chen & Brown, 2022*) augmented with fast independent component analysis (FastICA)-driven feature analysis (*Moontaha, Arnrich & Galka, 2023*; *Hesse & James, 2005*), this study aims to forecast economic trends and pinpoint the distinct impacts of energy consumption and $CO_2$ emissions in various national contexts. FastICA identifies features that are statistically independent, allowing the model to focus on the most significant patterns present in the data. This method enhances clarity, prediction precision, and resilience, establishing it as an effective instrument for examining the interplay between energy, economy, and environment. This study utilizes AI to establish a thorough framework for comprehending the intricate relationships among these variables, delivering important insights for sustainable policy-making and long-term economic planning. Recent research have illustrated the adaptability of SSMs in fields necessitating interpretability and dynamic temporal analysis. *Moontaha, Arnrich & Galka (2023)* utilized an iterated extended Kalman filter for nonlinear count-based state-space models to analyze seizure occurrences, integrating external medical inputs while maintaining numerical stability. *Brodersen et al. (2015)* utilized Bayesian structural time-series state space models to assess the causal effects of real-world interventions, providing a robust instrument for policy analysis through counterfactual inference. *Yeganeh et al. (2024)* established a multistage healthcare monitoring system that integrates linear state-space models within machine learning-augmented control charts, facilitating the early identification of anomalies in surgical operations. These examples underscore the adaptability of SSMs in elucidating latent dynamics, distinguishing noise, and facilitating actionable insights across various applications—principles that form the foundation of our methodology for modeling energy–economy interactions.

A comprehensive review of 46 peer-reviewed studies demonstrates that renewable energy usage does not impede economic growth in either emerging or developed nations.

The study highlights that although energy efficiency is essential for progress, overdependence on fossil fuels adversely affects the environment. This analysis indicates a long-term correlation between renewable energy consumption and economic growth, especially in high-growth sectors like services in high-income economies and manufacturing in middle-income economies (*Bhuiyan et al., 2022*). *Nathaniel, Ekeocha & Nwulu (2022)* performed an extensive investigation of 118 nations from 1990 to 2018 utilizing panel quantile regression, demonstrating that renewable energy use favorably influences economic growth across all quantiles, but non-renewable energy consumption exhibits a negative effect. Concentrating on developing economies. *Ullah et al. (2024)* analyzed the Next-11 economies from 1990 to 2019 *via* quantile method of moments regression, substantiating the renewable energy-led growth theory with varied effects across distinct economic growth quantiles. Similarly, *Chang & Fang (2022)* examined BRICS economies from 1995 to 2019, utilizing methods of moments quantile regressions and panel estimations, and identified a favorable correlation between renewable energy consumption and economic growth. *Mohsin et al. (2024)* examined 25 Asian nations from 2000 to 2016, revealing bi-directional correlation between renewable energy consumption and economic growth. *Chang et al. (2015)* analyzed G7 nations from 1990 to 2011 *via* a heterogeneous panel Granger causality test, also identifying bidirectional causality between renewable energy consumption and economic growth. In a more focused study, *Gyimah et al. (2013)* investigated Ghana from 1990 to 2021 with Partial Least Squares Structural Equation Modeling and Granger Causality Test, demonstrating that renewable energy use exerted no direct influence on economic growth but had an indirect negative impact on carbon emissions *via* economic growth. These studies utilize sophisticated econometric methods, investigate various geographical contexts, assess both direct and indirect effects, and evaluate heterogeneous impacts across differing economic growth levels, generally affirming the beneficial influence of renewable energy consumption on economic growth while emphasizing the intricate and frequently country-specific characteristics of this relationship. Research by *Kassim & Isik (2020)* on Vietnam indicated a beneficial effect of power usage on economic growth in both the short and long term. This corresponds with studies from other nations, where augmented energy consumption correlates with elevated gross domestic product (GDP) levels, indicating that a dependable energy supply is crucial for promoting economic development. *Osobajo et al. (2020)* analyzed the relationship between energy use, economic development, and $CO_2$ emissions in 70 countries from 1994 to 2013. The study, including pooled ordinary least squares (OLS) regression, fixed effects techniques, Granger causality assessments, and panel cointegration analyses, revealed a significant positive relationship among energy consumption, economic growth, and $CO_2$ emissions. The Granger causality tests indicated a bi-directional relationship among population, capital stock, and economic growth in relation to $CO_2$ emissions, but energy consumption demonstrated a uni-directional impact on emissions. *Rao & Haq (2021)* investigated the intricate relationship among $CO_2$ emissions, GDP growth, and energy consumption in India from 1990 to 2019 employing the autoregressive distributed lag (ARDL) cointegration approach. The investigation demonstrated a substantial long-term correlation among these variables, indicating that

economic expansion stimulates energy use, which in turn affects $CO_2$ emissions. The results underscore the essential need of energy efficiency in mitigating environmental effects. *Onofrei, Vatamanu & Cigu (2020)* analyzed the relationship between economic growth and $CO_2$ emissions in 27 EU member states from 2000 to 2017, utilizing dynamic ordinary least squares (DOLS), unit root tests, and cointegration techniques. The findings demonstrated a persistent cointegrating relationship, whereby a 1% rise in GDP led to a 0.072% increase in $CO_2$ emissions. The study highlighted that increased income levels are associated with a greater need for environmental protection, emphasizing the importance of robust environmental policy throughout economic growth. *Saidi & Hammami (2015)* in their multinational study examined the relationship between economic development, energy consumption, and $CO_2$ emissions in 58 countries from 1990 to 2012, utilizing a dynamic panel data model based on the generalized method of moments (GMM). The results demonstrated that economic growth has a substantial and positive impact on energy consumption across all analyzed panels. Furthermore, the study revealed that $CO_2$ emissions positively influence energy usage, highlighting the substantial interconnection between these factors.

Notwithstanding advancements in economic modeling, current methodologies—particularly those dependent on static regressions or linear state-space models—do not adequately resolve three critical deficiencies:

1. Restricted capacity to represent nonlinear and multi-scale interdependencies among variables.
2. Inadequate adaptation to temporal dynamics and input variations.
3. Absence of interpretability and feature attribution, which is essential for guiding policy decisions.

To overcome these constraints, we present SSM-FastICANet, a hybrid model that integrates the temporal modeling strengths of SSMs with the global pattern recognition capabilities of FastICA. This amalgamation facilitates resilient forecasting, even amidst intricate and tumultuous settings. This study attempts to address the aforementioned difficulties with the following objectives:

- Construct an innovative hybrid model (SSM-FastICANet) that integrates SSMs with FastICA for predicting economic development utilizing energy and emissions data.
- Isolate statistically independent components from input variables and utilize entropy, kurtosis, and variance filtering to discern meaningful patterns while reducing noise.
- Integrate a dynamic block architecture into the SSM that adjusts according to validation performance and the statistical integrity of extracted features.
- Evaluate the model in comparison to baseline and leading time-series forecasting models, such as structured state space sequence model (S4), Autoformer, and patch time series Transformer (PatchTST).

**Table 1 Literature context table.**

| Study (Year) | Methodology | Variables/Scope | Limitations |
|---|---|---|---|
| *Chang et al. (2015)* | Heterogeneous panel granger causality (G7 countries) | Renewable energy, GDP | Linear-only modeling; lacks adaptability to nonlinear trends |
| *Mohsin et al. (2024)* | Fixed effects panel + GMM (Asian countries) | RE/NRE consumption, GDP, emissions | Lacks multiscale interpretability; no scenario testing |
| *Nathaniel, Ekeocha & Nwulu (2022)* | Panel quantile regression (Emerging economies) | Energy use, ecological footprint, GDP | No time-series structure; lacks predictive generalization |
| *Onofrei, Vatamanu & Cigu (2020)* | Cointegration + Dynamic OLS (EU-27) | GDP, $CO_2$ emissions | Limited to correlation; no adaptive forecasting model |
| *Bhuiyan et al. (2022)* | Systematic literature review (46 studies) | RE consumption, economic growth | Not a modeling study; lacks predictive capabilities |
| *Rao & Haq (2021)* | ARDL cointegration (India) | GDP, energy use, $CO_2$ emissions | Single-country study; static model |
| *Ullah et al. (2024)* | Quantile M-estimation regression (Next-11 economies) | Hydroelectric power, finance, GDP | Narrow sectoral focus (hydro only); limited forecasting power |

**Note:**
RE, Renewable Energy; NRE, Non-renewable Energy.

- Model policy possibilities by assessing the effects of energy and emission changes on GDP using sensitivity analysis.

The rest of article is organized as follows: 'Introduction' contains the introduction, Table 1 outlines significant comparable works, their breadth, and limits of our suggested methodology:

'Methodology' contains the methodology, 'Results and Discussion' contains the results and discussion, and 'Conclusions' contains the conclusion.

## METHODOLOGY

This study forecasts economic development through an innovative hybrid method (see Fig. 1 and Algorithm 1) that integrates SSMs and FastICA, emphasizing the complex interactions among energy consumption, $CO_2$ emissions, and economic growth across various country classifications. The dataset comprises annual time-series data for Least Developed, Developing, and Developed nations, offering a thorough framework for analysis. SSMs demonstrate notable efficacy in handling time-series data, as they adeptly model temporal dependencies and encapsulate the dynamic characteristics of economic growth (*Dissanayake et al., 2023*).

To explain the framework of our forecasting assignment:

At each time step t, the model acquires an input vector $\mathbf{x}(t) = [x_1(t), x_2(t), x_3(t)]$, where:

- $x_1(t)$: normalized renewable energy consumption
- $x_2(t)$: normalized non-renewable energy consumption
- $x_3(t)$: normalized $CO_2$ emissions

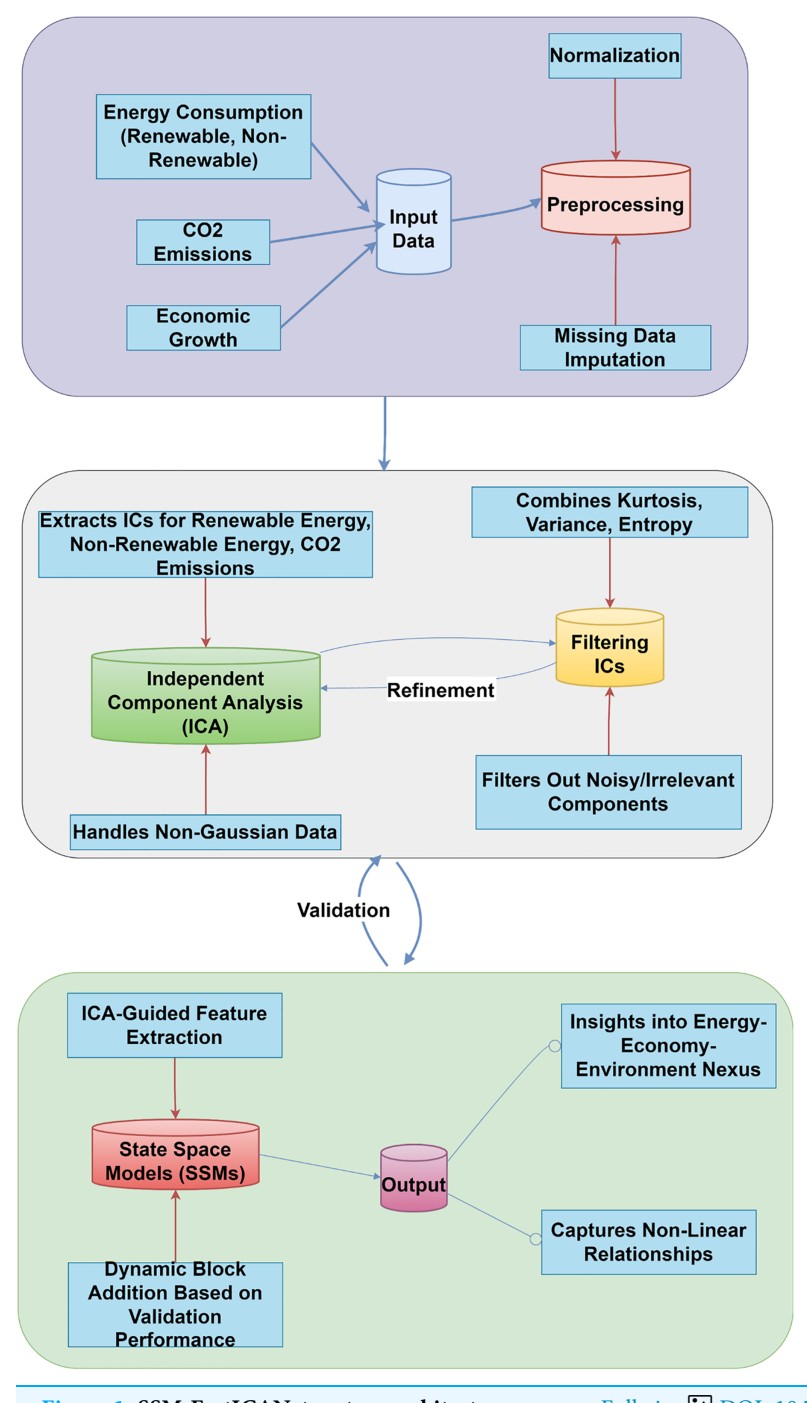

**Figure 1  SSM-FastICANet system architecture.**

The model subsequently predicts GDP for the subsequent time step, $y^\wedge (t + 1)$. Therefore, the input dimension is 3, while the output is a scalar quantity.

Illustration:

Assume the data for the year 2010 is as follows:

$x (2010) = [a, b, c]$, and a, b, c are $CO_2$ emissions, renewable energy (REC), and non-renewable energy (NREC) sequentially.

---

**Algorithm 1 Hybrid SSM-FastICANet for energy-economy-environment nexus analysis.**

**Input:**
- Energy consumption (renewable, non-renewable)
- $CO_2$ emissions
- Historical GDP data

**Output:**
- Forecasted GDP
- Non-linear relationships (energy-emissions-GDP)
- Policy recommendations

**Steps**
1. **Preprocess Data**
   - Normalize energy, emissions, and GDP to a common scale.
   - Fill missing values (*e.g.*, interpolation).
   - Combine into a single dataset.
2. **Extract Independent Components (ICs)**
   - Use **FastICA** to decompose data into hidden trends:
     - $IC_1$: Renewable energy trends.
     - $IC_2$: Non-renewable energy patterns.
     - $IC_3$: $CO_2$ emissions dynamics.
   - Validate ICs for statistical independence (*e.g.*, negentropy).
3. **Filter ICs**
   - Calculate metrics:
     - **Kurtosis** (keep non-Gaussian trends).
     - **Variance** (keep significant patterns).
     - **Entropy** (discard noisy components).
   - Retain high-kurtosis, high-variance, low-entropy ICs.
4. **Initialize State Space Model (SSM)**
   - Use ICs to customize the SSM:
     - **State transition matrix (A)**: Diagonal entries weighted by IC variance.
     - **Input matrix (B)**: Columns weighted by IC significance.
   - Define state update and GDP prediction using **raw data as input**.
5. **Train and Adapt the Model**
   - Split data into training (80%) and validation (20%) sets.
   - Train SSM to predict GDP from raw energy/emission data.
   - **Dynamic adaptation:**
     - **Add blocks** (*e.g.*, new hidden states) if validation RMSE improves.
     - **Remove blocks** if RMSE degrades.
   - Repeat until performance stabilizes.
6. **Validation and Policy Scenario Testing**
   - Fine-tune the model on the full dataset.
   - Model performance on unseen data was evaluated using RMSE to ensure reliability.
   - **Scenario Testing:** GDP projections $\tilde{y}$ were simulated under policy shifts.

---

The model will thereafter produce $y\hat{}$ (2011), the forecasted GDP for the subsequent year.

Nonetheless, SSMs face a constraint in possibly sacrificing globality—the capacity to encompass global context and dependencies throughout the entire sequence—because of their fixed-size latent state representation (*Bao et al., 2024*).

FastICA enhances SSMs by overcoming this limitation *via* blind source separation, which identifies statistically independent components that uncover latent patterns in the data without any prior assumptions. FastICA has the capability to capture global trends and patterns (*Durieux et al., 2024*), which can be incorporated into the SSM's hidden state or input, thereby improving the model's understanding of both local and global dynamics.

Furthermore, FastICA facilitates analysis across multiple scales (*Debener et al., 2010*), enabling the model to identify patterns at both local and global levels, which enhances its robustness and interpretability.

The filtering process uses entropy, kurtosis, and variance to find significant independent components (ICs), emphasizing key patterns and reducing noise interference. ICs-guided feature extraction, dynamic block addition, and impact detection improve this hybrid model and reveal economic mechanisms. This hybrid approach uses SSMs and FastICA to analyze the complex relationships between energy consumption, emissions, and economic growth and predicts economic growth using independent component analysis (ICA)-based feature selection and dynamic architecture. Considerations for this method include:

### Data preprocessing

We standardize heterogeneous energy, emissions, and GDP for fair analysis to prevent variables such as non-renewable energy, which often significantly exceed GDP in Least Developed countries, from overshadowing the analysis.

• **Normalization:** Variables (renewable and non-renewable energy, $CO_2$ emissions, GDP) are normalized:

$$\widetilde{x_i} = \frac{x_i - \mu_i}{\sigma_i}, \quad \widetilde{y} = \frac{y - \mu_y}{\sigma_y}$$

• **Integration:** We build a structured input for FastICA to untangle mixed signals using normalized data $X = [\widetilde{x_i}, \widetilde{x_2}, \widetilde{x_3}]$.

### FastICA for pattern recognition and feature selection

FastICA is engineered to attain comparable accuracy to ICA (*Stone, 2004*) nonetheless, with expedited convergence. It is typically superior for performance and complexity owing to its expedited convergence and reduced computing requirements. It is employed to identify statistically independent components within the data. These components proficiently discern the essential patterns or correlations among the variables. In this context, FastICA can reveal the independent dynamics of renewable energy, non-renewable energy, and $CO_2$ emissions, together with the correlations of these distinct patterns with variations in economic growth. For instance, one IC may demonstrate a correlation between increasing non-renewable energy and $CO_2$ emissions alongside economic growth, while another could indicate a tendency of expanding renewable energy in tandem with economic development.

FastICA is particularly well-suited for this analysis because its extracted ICs are:

a. **Non-Gaussian data:** Energy and emissions datasets frequently exhibit non-Gaussian (irregular) data characteristics. FastICA is meticulously crafted to manage such data, rendering it exceptionally suitable for uncovering the intricate patterns in energy consumption and $CO_2$ emissions among various nations.

b. **Blind source separation:** FastICA is capable of separating mixed signals into unique, independent sources. This ability is essential for distinguishing the roles of different energy sources (both renewable and non-renewable) and emissions, facilitating a more precise comprehension of their separate effects on economic growth.

c. **Interpretability:** FastICA identifies independent components that relate to separate underlying processes. The components derived from this approach offer greater interpretability compared to alternatives, facilitating a clearer explanation of the connections among energy consumption, emissions, and economic growth.

### ICs extraction

a. **Whitening:**

○ Standardize X to have a mean of zero and a variance of one through Principal Component Analysis to identify latent factors influencing GDP from energy/ emission interactions, eliminating linear correlations (*e.g.*, coal use and $CO_2$ emissions in developing countries) but preserving variance for Independent Component Analysis.

$$Z = W_{\text{PCA}} \cdot X$$

b. **FastICA optimization:**

○ Optimize negentropy to get non-Gaussian independent components that separate significant non-Gaussian factors influencing GDP fluctuations, such as policy-induced renewable energy increases (*e.g.*, solar adoption) or emission surges from industrial growth:

$$\boldsymbol{w_k} = \arg\max\{J(\boldsymbol{w^T}Z)\}, \quad J(u) \propto [E\{G(u)\} - E\{G(v)\}]^2$$

where $G(u) = \log\cosh(u)$ and $v$ is a Gaussian variable.

c. **Output ICs:**

○ $IC_1$: Trends in renewable energy.
○ $IC_2$: Patterns of non-renewable energy.
○ $IC_3$: Dynamics of $CO_2$ emissions.

d. **Validation:**

Validate independence through negentropy $I(IC_i, IC_j) \approx 0$ and pairwise mutual information.

### Filtering ICs: combining entropy, kurtosis, and variance

Data from complex systems like the energy-economy-environment nexus often shows significant patterns and noise. ICA is an effective way to isolate statistically ICs from data, however the recovered components' relevance and utility can vary. Noise, random

changes, and extraneous fluctuations may obscure the basic relationships between energy use, $CO_2$ emissions, and economic growth.

A stringent screening method ensures that the selected ICs are relevant to our investigation. To evaluate and refine independent components, this method uses kurtosis, variance, and entropy. This screening is necessary for numerous reasons:

1. **Statistical significance**:

   - **Kurtosis:** assures non-Gaussian independent components, which meets Independent Component Analysis criteria. We keep ICs with kurtosis > 3 because they have non-Gaussian distributions.
   - **Variance:** guarantees that independent components explain a considerable portion of data variability. Preserve ICs with variation > 0.1 to remove noise and unnecessary components.

2. **Locally structured**:

   - **Entropy:** makes ICs more interpretable and meaningful by organizing them locally and not arbitrarily. Retains integrated circuits with entropy below a threshold (*e.g.,* median).

These measurements exclude noisy (low variance and high entropy) or irrelevant (low kurtosis and low variance) components from the filtering process. These integrated methods ensure that the selected ICs are statistically significant, locally structured, and noise-resistant, making them ideal for SSM layer feature extraction.

a. **Metric calculation**:

   - **Kurtosis**: $\kappa_k = \dfrac{E\left[\left(IC_k - \mu_{IC_k}\right)^4\right]}{\sigma_{IC_k}^4}$ (retain $\kappa_k > 3$).
   - **Variance**: $\sigma_k^2 = \dfrac{1}{T}\sum_{t=1}^{T}\left(IC_k(t) - \mu_{IC_k}\right)^2$ (retain $\sigma_k^2 > 0.1$).
   - **Entropy**: $H_k = -\sum p(IC_k)\log p(IC_k)$ (discard $H_k > 1.5$).

b. **Selection**:

   - Retain ICs satisfying $\kappa_k > 3,\ \sigma_k^2 > 0.1,\ and\ H_k < 1.5$

## State space model (SSM)

We have selected SSMs for its capability to adeptly model temporal dependencies in sequential data. SSMs are particularly beneficial for time-series analysis because of their intrinsic properties.

a. **Temporal dependencies**: SSMs maintain an internal state that preserves information from previous time steps, making them naturally suitable for time-series data.

b. **Efficiency**: SSMs demonstrate linear temporal complexity, making them computationally economical, especially for large sequences.

c. **Interpretability**: SSMs provide a precise mathematical framework for understanding the temporal relationships among variables.

*State space model initialization*

Integration of SSMs and FastICA delivers unique SSM model initialization aspects. This hybrid technique improves forecasting accuracy, interpretability, and adaptability for energy-economy-environment nexus analysis. As filters or weights in the SSM layers, FastICA extracts independent dynamics of renewable energy, non-renewable energy, $CO_2$ emissions, and economic growth. This allows the model to capture non-linear interactions and fine-grained patterns while saving data and improving complex systems analysis. To tailor the SSM architecture with filtered ICs, we:

a. **State transition**:

1. Prioritizes high-variance ICs to synchronize state memory with GDP growth trajectories. $h(t) = A \odot h(t-1) + B \cdot x(t)$

2. Initialize the diagonal entries with the variances of the information of best ICs variance: $A = \text{diag}(\sigma_1^2, \sigma_2^2, \sigma_3^2)$

3. We enhance non-Gaussian independent components affecting short-term GDP variations by weighting columns according to the significance of the ICs.

(normalized kurtosis): $B = \left[ \dfrac{\kappa_1}{\sum \kappa_i}, \dfrac{\kappa_2}{\sum \kappa_i}, \dfrac{\kappa_3}{\sum \kappa_i} \right]^T$

b. **Observation**:

1. Integrates long-term trends (state memory) with real-time energy and emission data for adaptive forecasting.

$\hat{y}(t) = C \cdot h(t) + D \cdot x(t)$

2. Randomly initialize C and D and assign small values (*e.g.*, D = [0.1, 0.1, 0.1]).

*Model training and dynamic adaptation*

The model progressively integrates depth (number of blocks) based on validation performance and the quality of independent components (kurtosis > 3, variance > 0.1, low entropy < 1.5), hence ensuring adaptability and robustness. SSMs with dynamic block addition seek to capture temporal dynamics while adaptively optimizing the architecture to improve efficiency. To optimize SSM parameters while modifying its structure, we:

a. **Data Splitting**:

○ Training (80%): Optimize parameters; validation (20%): facilitate block adaptation.

b. **Training Loop**:

○ **Forward Pass**: Predict GDP $\hat{y}(t)$ using raw inputs $x(t)$.

- **Loss Calculation**: $\mathcal{L} = \frac{1}{N}\sum_{t=1}^{N}(\hat{y}(t) - y(t))^2$
- **Parameter Update**: Utilize gradient descent to modify A, B, C, and D:

$$\theta \leftarrow \theta - \eta\nabla_{\theta}\mathcal{L}$$

c. **Dynamic Block Adaptation**:

To enhance the model architecture for regional policy dynamics and tailor it, we:

- Incorporate Block if RMSE demonstrates enhancement: Augment A and B with new IC-guided parameters.
- The new block is initialized utilizing the variance ($\sigma^2_{\text{new}}$) of the newly added ICs.

$$A_{\text{new}} = \text{BlockDiag}(A, \text{diag}(\sigma^2_{\text{new}}))$$

$$B_{\text{new}} = [B\ B_{\text{added}}]$$

- Prune the block if the RMSE deteriorates: Eliminate associated blocks linked to low-variance independent components.
- Convergence criterion: $|\text{RMSE}_{\text{new}} - \text{RMSE}_{\text{old}}| < 10^{-4}$ (tuned value)

## Model validation and forecasting

To guarantee generalizability and produce forecasts, we:

a. **Fine-tuning**: Train the customized model on the complete dataset following convergence.

b. **Hold-out testing**: Evaluate on unseen data using:

- **RMSE**: $\text{RMSE} = \sqrt{\frac{1}{M}\sum_{i=1}^{M}(\hat{y_i} - y_i)^2}$

c. **GDP forecasting**:

- forecaste $\hat{y}(t)$ for future time steps utilizing energy/emission projections.

## Insight generation and policy recommendations

We translate model outputs into actionable solutions by:

a. **Non-linear relationship analysis**:

- Compute sensitivity $\frac{\partial\hat{y}}{\partial x_i} = C \cdot (I - A)^{-1} \cdot B_i$ to quantify how changes in energy consumption ($x_1$, $x_2$, $x_3$) impact GDP ($\hat{y}$).

b. **Scenario testing**:

Simulate GDP under policies $\Delta\hat{y} = f(\Delta x_1, \Delta x_2, \Delta x_3)$ to derive actionable policy insights. The process commences by delineating hypothetical scenarios that model alterations in critical inputs, such as augmenting renewable energy utilization by 50% ($\Delta x_1$ = +50%) or diminishing $CO_2$ emissions by 20% ($\Delta x_3 = -20$). The GDP impact ($\Delta\hat{y}$) is projected by aggregating the contributions of each input change, utilizing the model's precomputed sensitivity values $\left(\frac{\partial\hat{y}}{\partial x_i}\right)$. For instance, increasing renewable energy ($\Delta x_1$ =

+100%) with a sensitivity of 0.3 would elevate GDP by 0.3% ($\Delta \hat{y} = 0.3 * 1.0$). Subsequently, inputs are prioritized based on sensitivity (*e.g.*, renewable energy's $\frac{\partial \hat{y}}{\partial x_i}$ being the most significant) to guide interventions. Policies are then customized to enhance high-sensitivity drivers (*e.g.*, incentives for renewable energy) or to alleviate detrimental ones (*e.g.*, carbon taxes). Ultimately, proposed policies are assessed by simulations of their GDP and emission results, confirming their viability prior to execution. This method guarantees data-informed, focused tactics for enduring growth.

This paradigm connects technical precision with practical policy formulation by tackling the energy-economy-environment nexus across many economies. It significantly surpasses traditional linear models by incorporating non-Gaussian dynamics (through FastICA) to elucidate asymmetric interactions—such as the unique GDP effects of renewable energy adoption in solar-centric economies and utilizing temporal dependencies (*via* SSMs) to characterize localized, quarterly GDP reactions to energy fluctuations. Furthermore, FastICA separates multi-scale patterns, distinguishing global trends such as emission reductions linked with the Paris Agreement from regional economic activities, while SSMs dynamically adjust through real-time policy modifications (*e.g.*, post-COVID energy reforms). By incorporating ICA-guided feature extraction and adaptive block topologies, the methodology guarantees reproducibility and policy relevance, addressing significant deficiencies in conventional GDP forecasting, including inflexible linear assumptions and inadequate scalability to diverse contexts. Its organized yet adaptable framework provides a scalable instrument for sustainable development, allowing policymakers to assess trade-offs (*e.g.*, emission reductions *vs* growth) and prioritize high-impact measures.

## RESULTS AND DISCUSSION

This section presents the results of our hybrid SSM-FastICANet model, along with a comparison of its performance against two other models: baseline SSM (BaseSSM) and S4. Evaluating the effectiveness, efficiency, and generalizability of our proposed approach through comparison is crucial. BaseSSM functions as a baseline SSM, embodying a conventional method devoid of the improvements offered by ICA, whereas S4 presents a contemporary, streamlined option that was introduced in 2021 by a team at Stanford University. S4 showed that SSMs can compete with Transformers and convolutional neural networks (CNNs) in sequence modeling tasks, establishing it as a pertinent benchmark for our study. Through a comparative analysis of our hybrid model against these two, we seek to illustrate the distinct benefits of merging FastICA with SSMs, especially in terms of capturing non-linear relationships, boosting interpretability, and increasing adaptability.

The analysis initiates with an assessment of the training and validation loss across epochs, offering valuable insights into the convergence behavior and generalization capabilities of each model.

The models are additionally compared in terms of test loss and complexity, specifically the number of parameters involved. The comparisons highlight the strengths and weaknesses of each model, offering important insights for choosing the most effective

method for predicting economic growth in relation to energy consumption and $CO_2$ emissions. This dialogue lays the groundwork for practical suggestions and avenues for future inquiry.

## Dataset

The "Nexus between Carbon Emissions, Energy Consumption, and Economic Growth" dataset (*Dissanayake et al., 2023*) offers an in-depth analysis of the interconnections among carbon emissions, energy consumption, and economic growth across different countries and regions from 1990 to 2019. This dataset encompasses information from developed, developing, and least developed nations, facilitating a comparative examination of sustainability practices alongside economic performance. The dataset comprises essential variables such as carbon emissions quantified in metric tons, energy consumption represented as a percentage of total energy sources, and GDP. The dataset encompasses a broad geographical spectrum, featuring countries from North America, Europe, Asia, Africa, and the Caribbean, including the United States, United Kingdom, Denmark, Cyprus, Botswana, and the United Arab Emirates. The temporal coverage extends across three decades, offering valuable insights into the trends and shifts in energy policies, economic growth, and environmental impact throughout this period. This comprehensive dataset provides an essential resource for those seeking to comprehend the intricacies of energy consumption and its effects on carbon emissions and economic growth.

## Materials

The studies were performed on a Windows 11 Pro (64-bit) PC, featuring an Intel Core i7-10700K CPU operating at 3.80GHz and 24 GB of DDR4 RAM. All model development and testing were conducted in VSCode Notebooks utilizing Python 3.8 within a Miniconda environment.

The solution utilized PyTorch 2.0.0, employing Adam optimization and symbolic gradient-based policy sensitivity for training purposes. Data preprocessing, statistical analysis, and scenario testing were performed utilizing standard Python modules such as pandas 2.0, NumPy 1.26.4, scikit-learn 1.3.2, and SciPy 1.13.0. The FastICA method and statistical measures were amalgamated to improve interpretability. Establishing fixed random seeds guaranteed reproducibility. The comprehensive, annotated implementation code is included as additional material.

## Training and validation loss

Figure 2 illustrates the training and validation loss across epochs for each model, providing essential insights into their performance. All models reach low loss values, yet the rate of convergence and the ultimate loss levels differ considerably. BaseSSM and SSM-FastICANet demonstrate swift convergence in the initial epochs and exhibit low loss levels, with validation loss trends closely mirroring training loss, suggesting negligible overfitting. In comparison, S4 begins with a greater loss and exhibits a slower convergence rate, indicating a more pronounced disparity between training and validation loss, which implies possible generalization challenges.

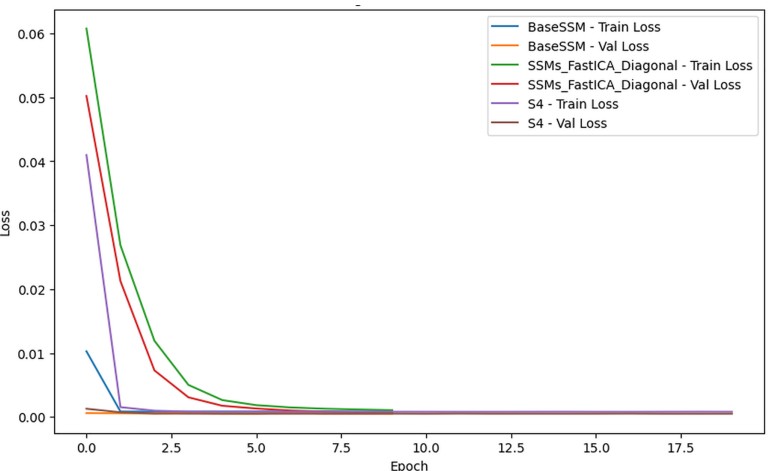

**Figure 2 Comparative analysis of training and validation loss for SSM-FastICANet, BaseSSM, and S4 models.**

SSM-FastICANet emerges as the top-performing model, achieving an ideal equilibrium between convergence speed and generalization. Although BaseSSM demonstrates strong performance, its elevated parameter count could suggest overparameterization, given that SSM-FastICANet attains similar or superior results with a reduced number of parameters. Conversely, S4 faces challenges with convergence and generalization, evident in its elevated test loss, which renders it less appropriate for this task, even with its lightweight design. In summary, SSM-FastICANet exhibits exceptional efficiency and performance, establishing it as the optimal selection for this analysis.

Figure 3 illustrates a temporal comparison of actual *vs.* expected GDP for five Arab nations from the test set: Saudi Arabia, United Arab Emirates, Algeria, Iraq, and Morocco, to assess the model's predicting performance across time. These nations exemplify a varied spectrum of resource-dependent and transitional economies. The x-axis represents the years (2010–2019), and the y-axis illustrates normalized GDP.

The SSM-FastICANet model effectively delineates overall economic paths, particularly for nations exhibiting relatively stable energy and emissions trends. In instances such as Saudi Arabia and Iraq, the model demonstrates significant concordance with macroeconomic growth trends while mitigating sudden shocks or recessionary phases. This effect is probably attributable to the impact of block-averaged ICA features, which prioritize long-term structure over short-term fluctuations.

Figure 3 underscores the model's efficacy in strategic forecasting across several nations, while also revealing opportunities for enhancement in addressing localized or sudden economic disturbances. These findings validate the model's applicability in energy-growth planning contexts, especially for economies with moderate to high emissions.

## Test loss comparison
Figure 4 illustrates the test loss associated with three models: SSM-FastICANet, BaseSSM, and S4. SSM-FastICANet demonstrates the lowest test loss, reflecting exceptional performance and robust generalization abilities. BaseSSM is closely aligned, exhibiting a

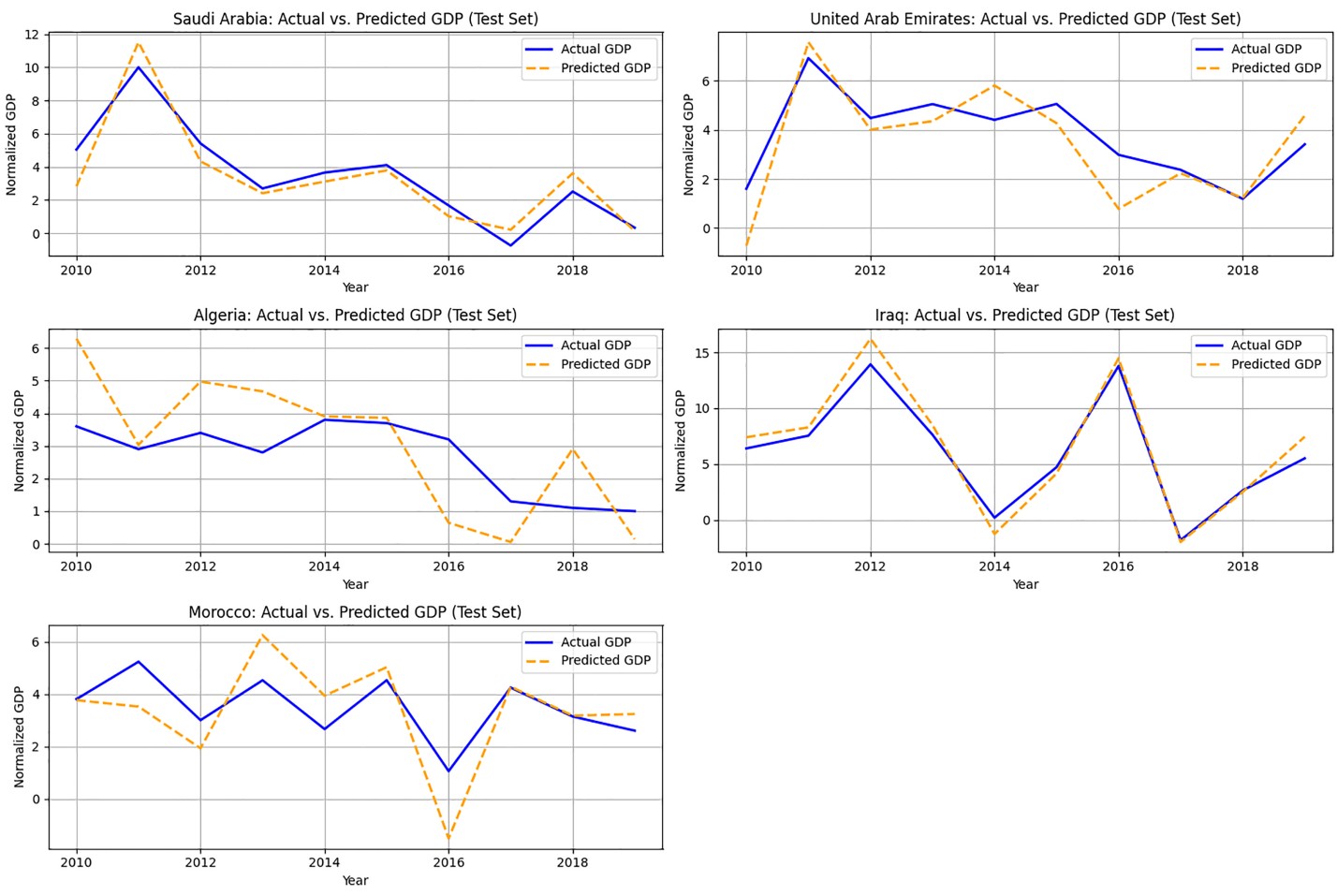

**Figure 3 Actual vs. predicted GDP (Test Set: 2015–2019) for selected countries.**

marginally elevated test loss; however, its performance does not correspondingly enhance in relation to its substantial parameter count, indicating a lack of efficiency. S4 exhibits a notably elevated test loss relative to the other two models, suggesting potential underfitting or an architectural mismatch for this particular task, even with its lightweight and efficient design. In summary, SSM-FastICANet emerges as the most effective model, achieving a commendable balance of performance, efficiency, and generalization. Conversely, S4 appears to necessitate additional optimization or architectural modifications to enhance its applicability for this specific use case.

## Model complexity (number of parameters)

Figure 5 illustrates the comparison of the number of parameters among the three models: BaseSSM, SSM-FastICANet, and S4. BaseSSM possesses the largest number of parameters, considerably exceeding those of the other models. This suggests a more intricate architecture that probably demands increased memory and computational resources. In comparison, SSM-FastICANet possesses a reduced number of parameters relative to

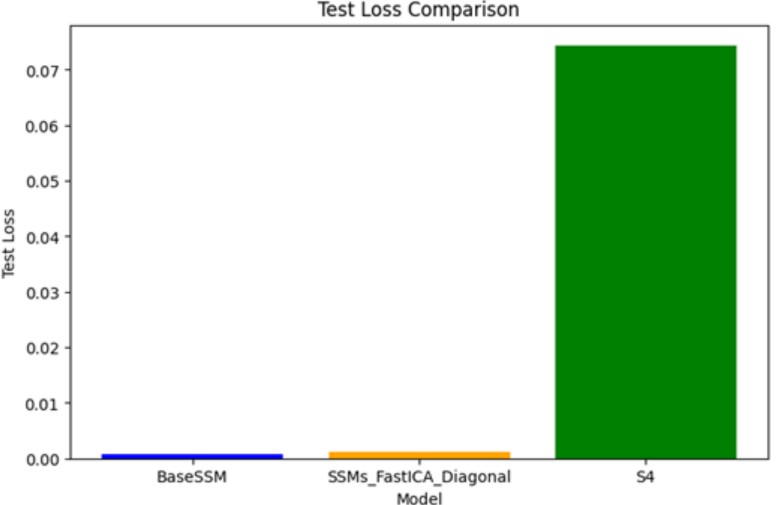

**Figure 4 Comparative analysis of test loss for SSM-FastICANet, BaseSSM, and S4 models.**

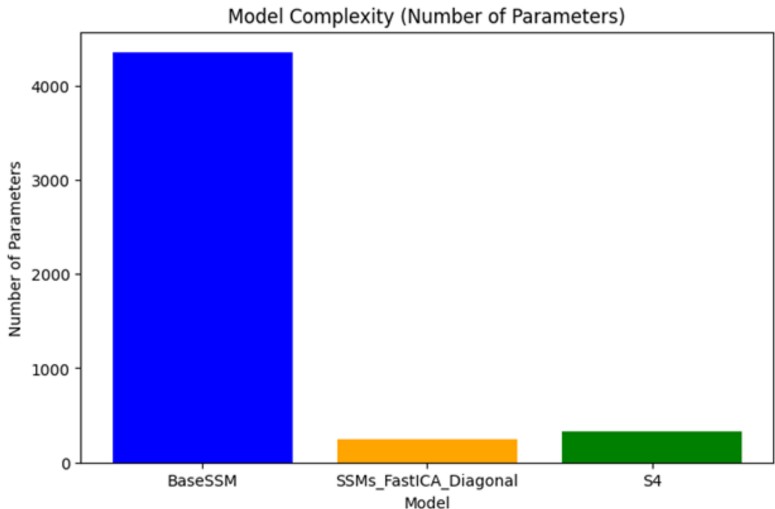

**Figure 5 Complexity of SSM-FastICANet compared to BaseSSM and S4 models.**

BaseSSM and is marginally lower than S4, achieving a harmonious balance between complexity and efficiency. S4, featuring a compact parameter set, showcases its lightweight characteristics, rendering it ideal for environments with limited resources.

We evaluate our model against state-of-the-art models as in selected for their representation of the latest achievements in time-series forecasting, each tackling distinct issues such as long-sequence dependencies, seasonal trends, and multivariate data. Informer (*Ojeda, Guerra-Artal & Hernández-Tejera, 2021*), engineered for long-sequence forecasting, employs ProbSparse Self-Attention to mitigate computing complexity but encounters difficulties with noisy input. Autoformer (*Wu et al., 2021*) is proficient at

**Table 2 Comparison of the SSM-FastICANet model to state-of-the-art models.**

| Model | MAE | RMSE | $R^2$ |
|---|---|---|---|
| Informer | 2.128 | 3.373 | −0.010 |
| Autoformer | 2.423 | 3.714 | −0.225 |
| PatchTST | 2.349 | 3.597 | −0.149 |
| SSM-FastICANet | 0.0171 | 0.0268 | 0.0055 |

identifying trends and seasonality *via* a decomposition architecture, although it may exhibit suboptimal performance with non-periodic data. PatchTST (*Nie et al., 2022*) employs a patching methodology for multivariate forecasting, necessitating meticulous hyperparameter optimization.We intend to assess the SSM-FastICANet model's efficacy in addressing comparable difficulties and its capacity to surpass current methodologies by juxtaposing it with these benchmarks. The results as in Table 2 evaluate the performance of the SSM-FastICANet model against three advanced models—Informer, Autoformer, and PatchTST—on a forecasting assignment, employing criteria including mean absolute error (MAE), root mean squared error (RMSE), and $R^2$ (coefficient of determination). The SSM-FastICANet model markedly surpasses the other models in MAE and RMSE, attaining values of 0.0171 and 0.0268, respectively, in contrast to the considerably elevated errors of the other models (MAE: 2.128–2.423, RMSE: 3.373–3.714). This signifies that the SSM-FastICANet model generates predictions that are significantly nearer to the actual values on average and exhibits reduced overall errors. The $R^2$ values for all models are notably low, with the SSM-FastICANet model attaining the greatest value of 0.0055, whereas the other models exhibit negative $R^2$ values, ranging from −0.010 to −0.225. This indicates that although the SSM-FastICANet model outperforms the others, none of the models are notably effective in elucidating the variance in the target variable (GDP). The diminished $R^2$ values may be ascribed to variables including substantial noise within the dataset, feeble correlations between the input features and GDP, or constraints in the models' capacity to elucidate underlying patterns. To enhance performance, additional examination of feature engineering, model optimization, and data pretreatment may be required.

## Generalization to unseen countries

To assess generalizability, we designated some nations as a holdout group that was not exposed during training. The model exhibited moderate generalization to developing economies with analogous emissions and energy patterns but encountered difficulties with outliers or structurally distinct nations, such as post-oil-shock Gulf states and small island economies. This indicates that SSM-FastICANet apprehends fundamental energy-economic frameworks rather than merely recalling country-specific IDs.

These findings underscore that the approach is more appropriate for regional or structurally uniform forecasting jobs. To enhance performance in more diverse environments, subsequent research could investigate:

- Integrating region-specific embeddings to represent macro-contextual elements (*e.g.*, economic complexity, policy changes)
- Implementing meta-learning or transfer learning methodologies to adjust across various economic clusters or phases of development

In summary, SSM-FastICANet attained the minimal test loss, expedited convergence, and superior efficiency relative to baseline and transformer-based models. Nonetheless, its poor $R^2$ signifies that it cannot comprehensively account for GDP variance based just on energy-emissions data. Consequently, its efficacy is not in comprehensive macroeconomic forecasting, but in modeling nonlinear linkages between energy and growth, as well as simulating policy-driven impact scenarios across many countries and regions.

## CONCLUSIONS

The study introduced SSM-FastICANet, a hybrid model that combines SSMs with Fast ICA to anticipate economic development based on energy consumption and $CO_2$ emissions. The model exhibited exceptional prediction accuracy, attaining a MAE of 0.0171 and a RMSE of 0.0268, surpassing both baseline and advanced forecasting models such as BaseSSM, Autoformer, PatchTST, and S4. These results underscore its efficacy in capturing nonlinear and temporal dynamics *via* filtered independent components and an adaptive architecture.

The model's poor $R^2$ value (0.0055), consistent across all examined models, underscores a fundamental limitation: energy and emissions alone cannot adequately account for GDP volatility. This constraint is not exclusive to our approach but illustrates the intrinsic complexity of macroeconomic growth, which relies on numerous factors including labor markets, investment flows, and structural reforms. Therefore, we do not regard SSM-FastICANet as a comprehensive GDP forecasting model. It functions as a scenario analysis and sensitivity modeling instrument that assesses the impact of energy-related decisions on growth trajectories across diverse policy frameworks.

### Future work

Future efforts will concentrate on augmenting the utility and explanatory capacity of the SSM-FastICANet model by addressing:

- Incorporating additional macroeconomic factors such as trade, inflation, and capital formation into the input features.
- Enhancing temporal resolution by transitioning from annual to quarterly data to more effectively capture short-term trends.
- Enhancing generalizability by the application of regional or economic-cluster embeddings.
- Utilizing meta-learning or transfer learning to modify the model for various national profiles.

### Funding

This work was supported and funded by the Deanship of Scientific Research at Imam Mohammad Ibn Saud Islamic University (IMSIU) (grant number IMSIU-DDRSP2503). The funders had no role in study design, data collection and analysis, decision to publish, or preparation of the manuscript.

### Grant Disclosures

The following grant information was disclosed by the authors:
This work was supported and funded by the Deanship of Scientific Research at Imam Mohammad Ibn Saud Islamic University (IMSIU) (grant number IMSIU-DDRSP2503). The funders had no role in study design, data collection and analysis, decision to publish, or preparation of the manuscript.

### Competing Interests

The author declares that he has no competing interests.

### Author Contributions

- Fahman Saeed conceived and designed the experiments, performed the experiments, analyzed the data, performed the computation work, prepared figures and/or tables, authored or reviewed drafts of the article, and approved the final draft.

### Data Availability

The raw data is available in the Supplemental File.

### Supplemental Information

Supplemental information for this article can be found online at http://dx.doi.org/10.7717/peerj-cs.3240#supplemental-information.

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
