# Peer review of "SSM-FastICANet: a hybrid state space and FastICA model for economic growth forecasting in energy-economy-environment systems"

_PeerJ Computer Science, doi:10.7717/peerj-cs.3240_

## Round 0.1 · original submission · Major Revisions

Reviewers 1 and 2 have identified some important issues that must be addressed in detail.

·

Basic reporting

As the state space model is utilized in this manuscript, I believe it offers novel contributions and has the potential to be accepted. However, at this stage, both major and minor revisions are necessary before it can be considered for publication.

It is recommended that the authors include a literature review table at the end of the Introduction to provide a clear overview of related works. Following that, the authors should clearly articulate the research motivations and highlight the existing knowledge gaps that this study aims to address.

The data is vague for me. Plot the time series of your data for the train and test set. The xlable should be the data time of the data. The manuscript would benefit from additional figures that provide a clearer understanding of the real data. Specifically, please include visualizations that illustrate the structure and characteristics of the training and testing datasets. This will help readers better understand the data distribution and how it is used in the modeling process.

The lag and prediction horizon used in the model are not clearly explained. At each time step, how many input data points (dimensions) are fed into the model? What is the dimension of the output? Please clarify these points in the manuscript. Additionally, provide a numerical example using a specific sample to illustrate the input and output dimensions more clearly.

The manuscript mentions a comparison with the Autoformer model. Could you please clarify whether you trained the Autoformer model from scratch or used a pre-trained version? Additionally, I recommend including a comparison with the Chronos model, as it is also a relevant and competitive approach for time series forecasting. In general, the performance comparison presented in the manuscript is not sufficiently rigorous. Please clarify how the parameters of the baseline models (competitors) were selected or tuned. Additionally, I recommend including LSTM as another competitive model in the comparison, as it is widely used and effective for time series prediction tasks.

I recommend that the authors plot the predicted values (yhat) alongside the actual values (y) as time series. The x-axis should represent time, allowing readers to visually compare the real and predicted values over the same time intervals. This will help demonstrate the model’s performance more clearly.

For the steps outlined in Algorithm 1, please specify the exact functions and Python packages used to implement each step. This will enhance the reproducibility of your work and provide clarity for readers who may wish to replicate or build upon your methodology.

Please consider attaching the implementation code for the baseline (competitor) models used in your study. To improve clarity and reproducibility, include comments within the code explaining each part, particularly how to run and evaluate the competitor models. This will help readers better understand your experimental setup and comparisons.

I believe the state space model is a powerful and versatile approach for time series analysis. It has demonstrated effectiveness across a wide range of applications, including healthcare, finance, and other domains. I recommend that the authors briefly discuss the broader applicability of state space models and their relevance in real-world scenarios. To support this discussion, please consider citing the following papers:
https://www.sciencedirect.com/science/article/pii/S093336572400068X
https://www.mdpi.com/1099-4300/25/10/1372
https://www.tandfonline.com/doi/abs/10.1080/10705511.2018.1502043

Experimental design

previous comments

Validity of the findings

good

Additional comments

no

Reviewer 2 ·

Basic reporting

This paper proposes a novel hybrid model, SSM-FastICANet, that integrates State Space Models with Fast Independent Component Analysis (FastICA) to forecast economic growth based on energy consumption and CO₂ emissions data.
The paper is well-structured, methodologically sound, and introduces an innovative hybrid modeling approach that contributes meaningfully to the literature on economic forecasting within the energy-economy-environment nexus.

I recommend that the paper be published, contingent upon the authors implementing the following revisions.

Comments to authors

a) The introduction should more clearly articulate the forecasting challenges in the energy-economy-environment nexus that existing models fail to address. Within 4-5 sentences.

b) Present a dedicated paragraph that summarizes how the proposed SSM-FastICANet addresses limitations in traditional models such as econometric regressions or linear SSMs. Within 5 sentences

Experimental design

Also:

c) Consider presenting the study’s main objectives in bullet points to increase readability and clarity. Around 4-5 bullet points. More clearly.

Validity of the findings

Furthermore;

d) More directly connect the use of FastICA and dynamic SSMs to the practical challenges of interpreting multi-scale, nonlinear data in the policy domain.

e) While the CPU and software stack are listed, also include versions of PyTorch, Scikit-learn, and libraries used for FastICA and preprocessing.

f) Mention whether the model can generalize to countries not seen during training, and discuss any observed issues in that regard. In a few sentences.

g) The future-work section, which is in the end of the paper, needs to be enriched.

Additional comments

The paper is in good standing. In my opinion the authors have done significant work. Looking forward to the revised version.

Reviewer 3 ·

Basic reporting

The article covers an important topic and is written in a style that’s generally clear and easy to follow. Still, there are a few reporting issues that should be addressed to help readers and future users of your method. First, the paper should include a direct link to the exact version of the dataset used, along with some practical guidance for accessing or preparing the data. Clarifying your decisions for a few steps, such as how you chose parameter thresholds for filtering components or how missing values were handled, may also be beneficial. The results and discussion might be clarified by tightening up certain sections that reiterate the same topics. Also, I suggest stating the approach's limits more clearly throughout the text rather than simply at the conclusion. Lastly, even if the code is provided, setting a clear random seed at the start for repeatability, listing software requirements, and including a quick "getting started" instruction would make it much easier for others to use.

Experimental design

Although the hybrid modeling approach is new and the experimental setting is ambitious, the extremely low R2 value raises severe concerns about the efficacy of the design. Despite its intricacy, the model is unable to account for the real variation in GDP across the sample when the R2 is so near zero. This undermines the practical value of the approach and calls into question whether the chosen features—energy use and emissions—are, on their own, sufficient for forecasting economic growth at this scale. The manuscript acknowledges this limitation in passing, but it deserves much more direct and critical attention. Readers need a transparent discussion about why the model underperforms in explanatory power, whether the problem lies in the predictors, data quality, or in the way the modeling task was framed, and what this means for the broader research question. Without this, the rest of the technical results risk feeling academic rather than actionable. I encourage the authors to squarely address this issue, reconsider the modeling assumptions if necessary, and be clear about what can and cannot be concluded from the experiments.

Validity of the findings

The model's limited explanatory power makes it challenging to interpret the results as meaningful or actionable. Even though your hybrid model routinely performs better than others on several error metrics, it is evident that there is no meaningful association between the predictors and GDP because the R2 values for all models are so low. This fundamentally limits the trustworthiness and significance of any reported performance gains, as well as the strength of the policy conclusions you draw. The manuscript should engage much more directly with this issue: What does it mean for your research question that none of the models explain GDP variance? Is the modeling technique unable to account for structural changes across nations, the use of annual data, or missing or poorly quantified variables? I suggest refocusing the conversation on these limitations while making it clear what the existing model can, if anything, show. The veracity of the results is still very much in doubt until this is accomplished.

Additional comments

Despite the study's technical detail and good intentions, the current findings do not support any significant scientific conclusions. The models—including your suggested hybrid—cannot account for the fluctuation in GDP, and the incredibly low R2 values preclude any claim to predictive power or novel insights for economic forecasting or policymaking. The absence of explanatory power means that, regardless of improvements over other models in terms of error metrics, the approach does not advance our understanding of the problem or provide evidence to guide real-world decisions. At present, the manuscript falls short of the standards required for publication in PeerJ Computer Science because it does not demonstrate that the proposed method can uncover any meaningful relationship or mechanism in the data. For this work to become publishable, it would need either a fundamental change in research design or a much more critical, evidence-driven explanation of why the model fails and what that implies for future research.

---

## Round 0.2 · accepted · Accept

The reviewers are satisfied with the recent changes proposed by the authors and therefore I can recommend this article for acceptance.

·

Basic reporting

-

Experimental design

-

Validity of the findings

-

Reviewer 2 ·

Basic reporting

Looks ok now.

Experimental design

Happy with the changes

Validity of the findings

Looks fine now